# SMS-I: Intelligent Security for Cyber–Physical Systems

Eva Maia *[iD], Norberto Sousa [iD], Nuno Oliveira [iD], Sinan Wannous [iD], Orlando Sousa and Isabel Praça [iD]

GECAD—Research Group on Intelligent Engineering and Computing for Advanced Innovation and Development, School of Engineering of the Polytechnic of Porto (ISEP), 4249-015 Porto, Portugal
* Correspondence: egm@isep.ipp.pt

**Abstract:** Critical infrastructures are an attractive target for attackers, mainly due to the catastrophic impact of these attacks on society. In addition, the cyber–physical nature of these infrastructures makes them more vulnerable to cyber–physical threats and makes the detection, investigation, and remediation of security attacks more difficult. Therefore, improving cyber–physical correlations, forensics investigations, and Incident response tasks is of paramount importance. This work describes the SMS-I tool that allows the improvement of these security aspects in critical infrastructures. Data from heterogeneous systems, over different time frames, are received and correlated. Both physical and logical security are unified and additional security details are analysed to find attack evidence. Different Artificial Intelligence (AI) methodologies are used to process and analyse the multi-dimensional data exploring the temporal correlation between cyber and physical Alerts and going beyond traditional techniques to detect unusual Events, and then find evidence of attacks. SMS-I's Intelligent Dashboard supports decision makers in a deep analysis of how the breaches and the assets were explored and compromised. It assists and facilitates the security analysts using graphical dashboards and Alert classification suggestions. Therefore, they can more easily identify anomalous situations that can be related to possible Incident occurrences. Users can also explore information, with different levels of detail, including logical information and technical specifications. SMS-I also integrates with a scalable and open Security Incident Response Platform (TheHive) that enables the sharing of information about security Incidents and helps different organizations better understand threats and proactively defend their systems and networks.

**Keywords:** cyber–physical systems; digital forensics; cyber–physical systems forensics; machine learning; rule mining; security incident response

## 1. Introduction

Cyber–physical systems (CPS) combine the physical and cyber worlds, which allows an improvement of the entire operating environment by adding different promising capabilities to these environments [1]. Therefore, CPS are being used in several domains, including manufacturing processes, healthcare, transportation, and commercial and residential smart buildings [2]. For example, recently, several studies have been done to explore the full potential of CPS in the context of Industry 4.0 [3,4]. This can happen because CPS use and integrate different technologies, from software systems, networks, and sensors to hardware devices such as microcontrollers and actuators. However, this combination enabling interactions between cyber and physical components, not only brings new and more complex paths of attack but also increases the attack impact, since an event caused by a cyber component can have a huge impact on physical ones or vice-versa [5]. The connections between the physical systems and the critical software components are especially vulnerable, since with a cyber attack in these connections the attacker can manipulate, disrupt or power off the physical system [6]. Thus, beyond damage to cyber and physical components, a cyber–physical attack can also have major consequences that may include human deaths and injuries, infrastructure damages, loss of resources, and machine breakdowns or malfunctions. Furthermore, these damages can have an even greater impact

on critical infrastructure such as hospitals and airports. Stuxnet worm [7], the US power grid attack [8], German steel-mill Incident [9], the Ukrainian power grid Incident [10], and the recent Florida Water Treatment Plant [11] and Colonial Pipeline [12] attacks, are some examples of security attacks on CPS that have caused huge impacts on the normal operation of the systems.

After an attack, it is crucial to understand how it was performed, who did it and why it happened. This will help to understand which assets were compromised but also will allow the creation of defense mechanisms for future attacks [13]. For that, security analysts need to analyse and investigate several sources of information. In CPS, this investigation process becomes much wider and complex, due to the amount of components that need to be analysed. Not only software and hardware components need to be considered but also all interactions across all CPS. Several investigations have been done to develop tools to secure CPS as well as techniques and frameworks to evaluate CPS security; however, CPS forensic investigation area is still in its early stage. Mohamed et al. [14] reviewed examples of current research efforts in the field and the types of tools and methods proposed for CPS forensics. The authors also discussed some issues and challenges in the domain that need to be addressed. One of the issues pointed out was the need for data analytics tools to find correlations between digital and physical evidence. Furthermore, Fausto et al. [15] pointed out that finding complex attack patterns through the combination of physical and cyber Events is a very challenging task. Moreover, they stated that the correlation strategies of heterogeneous Events for security reasons, and the techniques and algorithms to exploit this correlation are still open issues.

Additionally, for a successful correlation of the security Events, it is essential to keep track of the currently handled Events. For that, cybersecurity teams typically use ticketing systems that allow the follow up of the event for analysis, after the reporting, and until closure. However, due to the complexity of modern attacks, increasingly multi-step, the Events handled can be part of a larger attack that spans different parts of systems. Thus, the information crucial to detecting such attacks is often distributed in time and space, which makes detection difficult. Hence, an important feature of these systems is the collaboration among the security professionals, such as Security Operations Center (SOC) and Computer Emergency Response Team (CERT) security analysts, with diverse knowledge, skills and experience, to improve the quality of their investigation. Moreover, collaboration is important not only between the security professionals of the same institution, but also between companies, sectors, and even countries to improve the exchange of information to prevent, mitigate and recover from cyber-attacks. Collaboration between these actors is crucial to restricting the spread of new attacks, particularly zero-day attacks. Sharing new vulnerabilities, attacks, breaches or any other type of information allows a proactive detection of these newly identified threats [16]. This way, the company, sector or even country under attack will benefit from the analysis and correlation actions previously defined by others to resolve the same or similar issues. Governments with their national cybernetic emergencies response team (CERT) or CSIRT are boosting this collaboration to provide support in information security Incidents to the government or corporate entities for the management of cybersecurity and cyberdefense. In addition, European regulatory directives [17] and technical recommendations [18] are promoting actions to ensure a high common level of network and information security across the Union, by developing technologies and procedures for sharing security information to combat modern attacks and mitigate their effects in a timely manner. [19] The aim is to work in a collaborative framework between the CERTs and CSIRTs of the governments that allow the share of information at the taxonomy level about vulnerabilities and reports to be interconnected, providing a large scale security situation awareness which is in turn critical to the overall security posture of an entire nation [20].

In this work, we describe the SMS-I tool, which deals with the analysis of data from heterogeneous systems over different time frames, correlates them to find evidence of the causes of an attack, and supports the definition of remediation measures in a collabora-

tive way. SMS-I was firstly designed in the scope of the SATIE project, which aimed to build a security Toolkit [21] in order to protect critical air transport infrastructure against combined cyber–physical threats by improving the cyber–physical correlations, forensics investigations and dynamic impact assessment at airports. However, SMS-I can work with data from any security CPS since it analyses additional security details, providing contextual and semantic data to identify causes for security events and threats. Furthermore, Machine Learning (ML) methodologies have been applied for outlier detection, exploring the temporal correlation between cyber and physical Alerts, going beyond traditional one-class algorithms, and considering ensemble methods to detect unusual events, taking into account its sequential nature, which may help to find evidence of attacks. An intelligent dashboard is also part of the SMS-I in order to support decision makers in a deep analysis of how the breaches and the assets were explored and compromised. SMS-I also integrates with a scalable and open Security Incident Response Platform (TheHive) that enables the sharing of information about security Incidents. This can make the difference to the organizations security, since this collaborative sharing of information can help different organizations better understand threats and proactively defend their systems and networks.

SMS-I can be easily extended with new modules that can increase its capabilities. Therefore, this work presents a more complete version of the SMS-I tool. A first draft was presented at [22], and a more complete version of this draft was presented at [23]. This work shows in more detail the capabilities presented in the previous works, but also introduces a new capability: the Incident response. Therefore, the main contributions of this paper are:

- detail the SMS-I tool capabilities. The different components of this investigation tool are fully described in this work, presenting its different features;
- present all the different experiments done regarding the SMS-I Machine Learning Engine. Some of these results are already presented in the previous papers; however, in this work, we detail all the work carried out and the results obtained;
- introduce the Incident response capability of SMS-I tool. This is a new SMS-I capability that promotes the sharing of information between organizations. The integration of this feature with TheHive is also detailed in this work;
- show SMS-I Intelligent dashboard in detail, highlighting the added value for the security analysts of each view;
- demonstrate the convenience and usefulness of the SMS-I tool in the decision-making process of security analysts, using a very simple and realistic example.

The remainder of this paper is organized as follows: in Section 2, we introduce the SMS-I architecture, and we briefly describe each component. The Machine Learning Engine is the heart of the SMS-I tool. Hence, Section 3 presents this SMS-I component with more detail. Section 4 describes the SMS-I intelligent dashboard, another important element of the SMS-I tool. The SMS-I Incident Response capability is detailed in Section 5. In the scope of SATIE project, the SMS-I tool was validated and demonstrated in three different airports. Section 6 briefly describes an example that shows the ability of SMS-I to support the security experts work. Finally, the conclusions are presented in Section 7.

## 2. SMS-I Tool Overview

SMS-I is a forensics investigation system that was initially designed to be part of the SATIE security Toolkit. However, as already mentioned, it can be part of any security system. To explain the integration of SMS-I in a security environment, we will use the SATIE example. Note that the referred SATIE systems can be easily replaced by any other similar security systems.

In the SATIE security environment, cyber and physical sensors are scattered across the whole airport's infrastructure, collecting vast amounts of Events related to the airport system's activity. These Events are sent to the Correlation Engine (CEngine), a pattern matching mechanism that contains expert written rules which are periodically reviewed and updated under a strict protocol, to possibly identify abnormal behaviour. When a set of Events trigger a specific rule, an Alert is originated and sent to the Incident Management

Portal (IMP). In the IMP, after investigating the Alert occurrence, the security operator classifies Alerts as either Incidents or not, triggering a security response. SMS-I tool inspects these Incident and Alert occurrences to provide a deeper analysis of an attack. For that, the system periodically fetches data from the CEngine and the IMP using HTTP(S) requests to obtain Alerts and Incidents generated by the SATIE Toolkit. These data are parsed into predefined formats and stored in specific indexes of the SMS-I Database. This is a crucial part of the SMS-I tool since it allows the system to keep track of the new data that is generated within the SATIE Environment. Then, the SMS-I ML Engine gets this new data and executes the ML models capable of determining, for each Alert, the probability of it being an Incident based on its own features, features of related Events and the features of other Alerts of a regarded time window (Incident Prediction). The employed models are expected to grow smarter over time with system usage. SMS-I ML engine also analyses these data to understand if the system already has remediation measures for the Incident that have occurred and, if not, supports the security analyst in its definition (Incident Response). Additionally, using the Association Rule Mining (ARM) Engine, the SMS-I ML Engine provides an API endpoint for executing rule mining algorithms on the SMS-I Database data according to a set of parameters specified in the request header (Association Rules). It retrieves the list of association rules to identify potential relationships between Alerts for a given timeframe.

The SMS-I Intelligent Dashboard provides a Graphical User Interface of all of these data that handle the interaction with the security analyst. It encapsulates Kibana dashboards and allows the operator to make use of several functionalities such as consulting Alert lists, performing filtering, mining new association rules, managing association rule base, and consulting Alert details. SMS-I also integrates with the TheHive Incident management tool that allows the collaborative investigation of Incidents. TheHIVE platform is a popular and recommended tool for the management of Incident cases [20]. It is tightly integrated with MISP (Malware Information Sharing Platform), which allows the exchange of information on information security Incidents, both internally and between other security teams. TheHive platform can be complemented with the Cortex engine to analyze the Incidents using advanced intelligence. An overview of the SMS-I architecture can be seen in Figure 1.

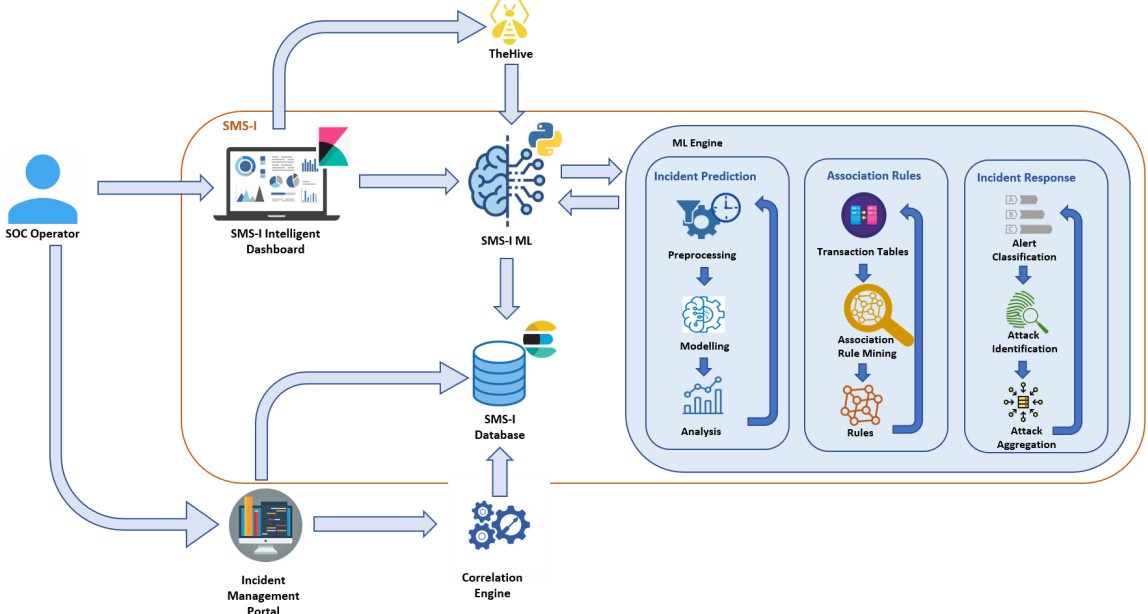

**Figure 1.** SMS-I architecture overview.

*2.1. SMS-I Integration*

In this section, SMS-I functionality and integration features are described. The SATIE system's security framework from the SMS-I perspective will be used to provide a better understanding of both SMS-I functionalities and its integration with other SATIE Tools such as the CEngine and the IMP. Note that this should be seen just as an example, and the referred tools can be easily replaced by other security tools, as already mentioned.

As a first step, it is crucial to formally define the fundamental business concepts—Events, Alerts, and Incidents—since they are constantly mentioned throughout the document:

- **Events** are discrete change of state or status of an Asset or group of Assets. They can have multiple heterogeneous sources and are categorized as either cyber or physical, depending on the system that originated them. They contain low-level information about the system's activity, such as network traffic or baggage handling system data. Specific Events may trigger Alerts.
- **Alerts** are notifications that a specific attack has been directed at an organization's information systems. They are triggered when abnormal activity is detected. They are usually related to several Events that have triggered security rules.
- An **Incident** results from the classification of Alerts by the SOC operator. They represent real identified threats to the system. Additionally, it has some sort of impact within the organization, which is described by its severity and completion level.

Unified Modeling Language (UML) and a combination of C4 Model [24] with 4+1 Architectural View Model [25] are used as a formalism to graphically represent software architecture from different views with different degrees of granularity. For example, the following diagram, Figure 2, provides a logic view of the SATIE security ecosystem without the SMS-I tool.

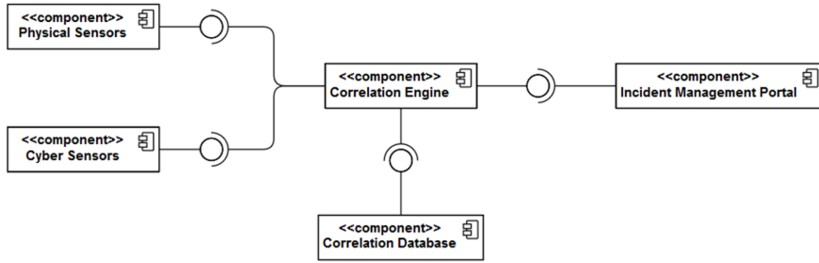

**Figure 2.** SATIE security ecosystem without SMS-I tool.

Different cyber and physical sensors present in the airport's infrastructure send a large amount of Events related to the airport system's activity. CEngine receives all these Events and stores them in the Correlation Database. When a set of Events triggers a specific rule of CEngine, an Alert is sent to the IMP to be analysed by a security expert and classified as an Incident or not, triggering a security response if needed.

SMS-I, as a forensics investigation system, will use an intelligent layer to help the security expert to inspect Incident and Alert occurrences. For that, the system periodically fetches data from the CEngine and the IMP, using HTTP(S) requests to obtain new Events, Alerts and Incidents generated by the SATIE Toolkit. These data are processed and stored in the Investigation Database of SMS-I, so it can be used by a web application to display several useful visualizations and by an ML Engine. The internal architecture of the SMS-I tool is described in greater detail in the next section. The following diagram, Figure 3, places SMS-I in the context of the SATIE solution as example of integration.

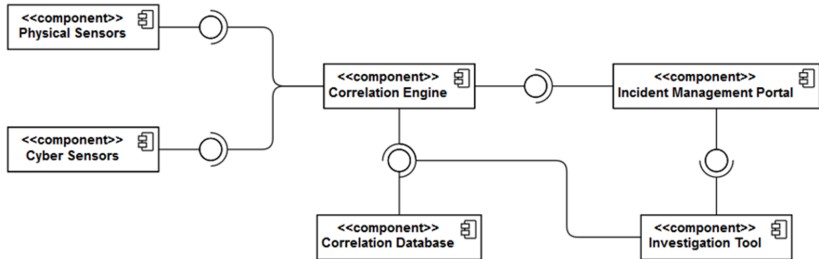

**Figure 3.** SATIE security ecosystem with SMS-I.

*2.2. SMS-I Internal Architecture*

SMS-I is a complex system with many different software requirements such as periodic data synchronization, Incident prediction and response computing with ML Engine, association rule mining, dashboard visualization and a series of other functionalities involving different lists and filters. To assure separation of concerns, modularity, and maintainability the system's architecture was designed with the Single Responsibility Principle (SRP) [26] in mind and inspired by a microservices-oriented architecture. Therefore, SMS-I is composed of multiple components with specific well-defined responsibilities. The internal architecture of the forensics investigation system is described in Figure 4.

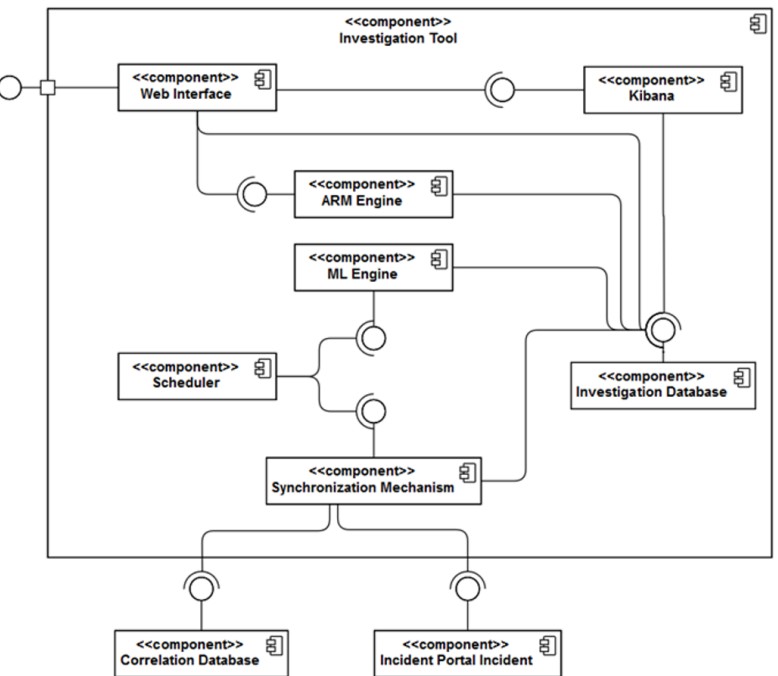

**Figure 4.** SMS-I architecture.

Each component of the SMS-I architecture can be described as follows:

- **Synchronization Mechanism:** It is the component responsible for acquiring new Events, Alerts and Incidents from the Correlation Engine and the Incident Management Portal, parsing them into predefined formats and storing them into specific indexes of the Investigation Database. The synchronization mechanism is one of the most critical processes of the SMS-I since it allows the system to keep track of the new data generated within the SATIE Environment. Additionally, as new Alerts are added to the database, they are also processed by the ML Engine. The synchronization process is represented in Figure 5.
- **ML Engine:** The ML Engine is responsible for executing the ML models capable of determining, for each Alert, the probability of it being an Incident based on its own

features, features of related Events and the features of other Alerts of a regarded time window. The employed models are expected to grow smarter over time with system usage. The ML engine also analyses the data received from an Incident response point of view, taking into account a collaborative approach and providing confidence scores over other related cases.

- **Scheduler:** The Scheduler performs the orchestration of both Synchronization Mechanism and ML Engine by triggering their execution by a configurable time constraint (e.g., every five minutes, every hour, every day).
- **ARM Engine:** The Association Rule Mining (ARM) Engine provides an API endpoint for executing rule mining algorithms on the Investigation Database data according to a set of parameters specified in the request header. It retrieves the list of generated rules.
- **Investigation Database:** It corresponds to an Elastic Search database that stores all system data—Events, Alerts, Incidents, ML probabilities and association rules.
- **Kibana:** It is part of the ELK Stack and can be described as an interface to the Investigation Database. It provides several methods to build interesting visualizations that are combined to produce intuitive and informative dashboards for inspecting the system's behaviour over time.
- **Web Application:** It provides a Graphical User Interface (GUI) that handles the interaction with the SOC operator. It encapsulates the Kibana dashboards and allows the operator to make use of several functionalities such as consulting Alert lists, performing filtration, mining new association rules, managing association rule base and consulting Alert details.

An Authentication module also grants authentication to the Web Application by matching user credentials with those stored in a shared LDAP server between all SATIE Tools. Lightweight directory access protocol (LDAP) is a protocol for accessing and maintaining data through directory servers often used for authentication and storing information about users, groups, and applications. This implementation allows every user to access every SATIE Tool with the same credentials.

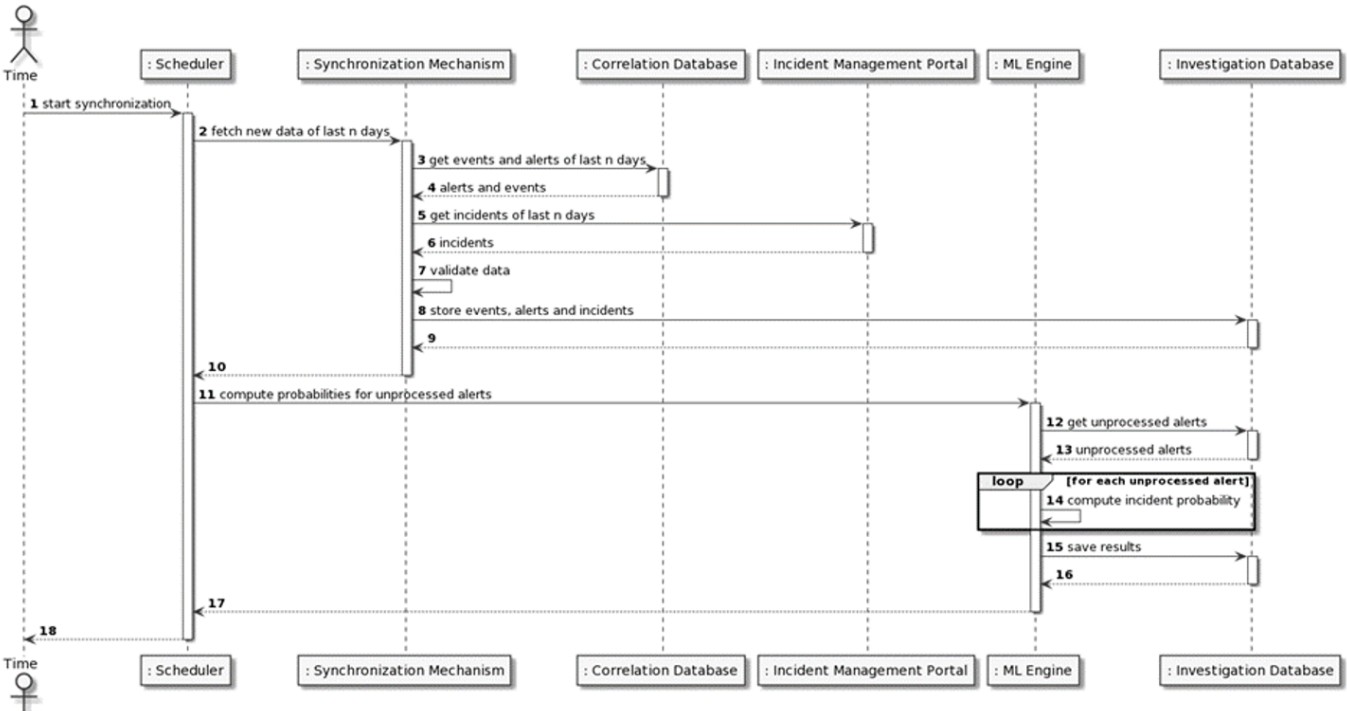

**Figure 5.** SMS-I Synchronization process.

## 3. SMS-I Machine Learning Engine

The ML methods present in the SMS-I can be categorized into three groups: Incident probability prediction, association rule mining and Incident response. For the first, supervised algorithms were trained on the sequential data of cyber and physical Alerts to predict the probability of a given Alert being an Incident based on previous occurrences. The second group of methods uses the same data to derive new correlation rules between Alerts that can be analysed to understand the complex pattern inherent to such data. The third group analyses the data to understand if the needed mitigation measures are already in place for the Incident reported. All these groups will be described in the following sections.

### 3.1. Incident Probabilities

There are many approaches for building ML models that can efficiently detect anomalies in time series data. To properly investigate and explore state of the art methods for such task, a study on public datasets was first performed. One of the difficulties of this study was to find an appropriate testbed for testing the employed methods performance. Currently, in the literature, there is not a huge amount of cyber–physical datasets being one of the most relevant the Secure Water Treatment (SWaT) dataset [27]. However, the physical data are too context-specific and there is no sufficient guarantees that a method able to safeguard a water treatment facility is going to exhibit the same kind of performance in the airport security domain, since they regard different physical sensors. The solution to this problem was to consider only the data from the network under study, which are more general and share many similarities between several domains, providing a better estimate of the model's performance. For example, the same kind of attacks, such as brute force and denial of service can be performed on many different networks to disrupt one or several services. Therefore, we decided to consider network intrusion detection datasets. And despite the lack of good and reliable datasets has been appointed in the literature as one of the main obstacles in intrusion detection research [28], some datasets were recently introduced to solve this issue, namely NSW-NB15 [29], CICIDS2017 [30] and CIDDS-001 [28]. From all the ones previously mentioned, CIDDS-001 was the one selected to be used for several reasons, such as the number of records, the recording period duration and the considered attack types. A comparison between the datasets mentioned above can be found in Table 1.

**Table 1.** Dataset comparison.

| Dataset | Year | Format | Count | Duration | Kind |
|---------|------|--------|-------|----------|------|
| NSW-NB15 | 2015 | packet, other | 2 M | 31 h | Emulated |
| CICIDS2017 | 2017 | uni. flow | 3.1 M | 5 Days | Emulated |
| CIDDS-001 | 2017 | uni. flow | 33 M | 28 Days | Emulated and Real |

Anomaly detection for the CIDDS-001 dataset, considering the AttackType label, was addressed using two different approaches: single-flow and multi-flow. The first regards individual flows as separate records and attempts to find differences between normal and attack related ones. The latter considers a given window of flows, performing an analysis on the entire data sequence to detect anomalies. For each approach three ML algorithms were experimented and compared: Random Forest (RF), Multi-player Perception (MLP) and Long-Short Term Memory (LSTM). In the next sections, we briefly describe this work. For more detail please see [31].

### 3.1.1. Incident Probabilities Testbed

The CIDDS-001 network traffic data are represented in unidirectional netflow format which, is a universal standard. The data were recorded for approximately four weeks from two different environments, an emulated small business environment, OpenStack, and External Server, which captured real traffic from the internet. The OpenStack environment includes several clients and servers, such as e-mail and web server. In this testbed, four

different types of attacks were performed: ping scans, port scans, brute forces and denial of service. The considered traffic data regards several features such as source and destination ports, source and destination IPs, communication protocol, number of transmitted bytes, number of transmitted packets, flow duration and TCP flags. Additionally, the data has three different labels, Class, AttackType and AttackDescription. For this investigation, the AttackType label was used since it provides a categorization of the different attacks that were performed. The considered algorithms were trained with this label so that they could recognise and distinguish the different attacks present in the testbed.

Random Forest (RF) is a supervised learning algorithm that uses an ensemble of decision trees, useful for classification or regression problems. Each decision tree that composes the "forest" reaches a prediction and the results of all of them is selected by majority voting or the average of outputs. By having multiple uncorrelated models for each of the trees, the possible individual errors of each one were diluted, relying on the "wisdom of the crowd" [32]. Another helpful model for classification and regression is a feed-forward neural network, Multilayer Perceptron (MLP). An MLP is a network of several layers of nodes, or neurons, each one with an activation function that maps the weighted inputs to the output of each node. Although feed forward means the data moves in only one direction, this model does benefit from back propagation during training, where the error between the prediction and the real value is fed back through the network to adjust the weights of each connection [33]. Due to the nature of the dataset used, a Long Short-Term Memory model were also employed. This neural network, unlike normal feed forward networks such as the previous example, has feedback connections. This allows it to process sequences of data such as network or Intrusion Detection System (IDS) [34].

### 3.1.2. TestBed Results

For evaluating and comparing the algorithms performance the dataset was split into three sets, training, validation and testing. The models were trained using the labelled data of the train set and their predictions were computed for the validation and testing set. By comparing these predictions with the real values several indicators of the methods performance can be calculated such as:

$$\text{Accuracy} = \frac{\text{Number of correct predictions}}{\text{Total number of predictions}}, \quad \text{Precision} = \frac{\text{Correct positives}}{\text{Total number of positives}},$$

$$\text{Recall} = \frac{\text{Correct positives}}{\text{Total number of positive samples}}, \quad \text{F1-score} = \frac{2 \times \text{Precision} \times \text{Recall}}{\text{Precision} + \text{Recall}},$$

$$\text{FPR} = \frac{\text{number of false positives}}{\text{total number of negatives}}.$$

Accuracy is biased towards the majority class, normal traffic, since it is obtained by dividing the number of correct predictions by the total number of observations. Hence, F1-score provides a better evaluation of an algorithm's performance since it is the harmonic mean of precision and recall. For the single-flow approach the obtained results are presented in Table 2.

**Table 2.** Results for the single-flow approach.

| Model | Accuracy | Precision | Recall | F1-Score | FPR |
|-------|----------|-----------|--------|----------|-----|
| LSTM | 99.91 | 98.37 | 71.40 | 74.23 | 00.05 |
| RF | 99.90 | 79.43 | 95.68 | 85.04 | 00.02 |
| MLP | 99.92 | 78.68 | 73.75 | 75.79 | 00.06 |

Analysing the results, it can be said that the best performing model was the RF with a F1-score of 85.04, it also exhibits lower recall in comparison to its value of precision. On the other hand, the LSTM has better precision with lower recall presenting an F1-score of 74.23. The MLP is quite balanced in terms of both metrics which resulted in an F1-score of 75.79, higher than the one of the LSTM. The RF also presents the lowest occurrence of false alarms, a FPR of 00.02 being arguably the best model for the single flow viewpoint.

For the multi-flow approach, the results are quite different. With the increase of the flow window size the results of the LSTM keep improving while the ones of RF and MLP decrease. Nevertheless, the RF for a window of 10 flows presents an F1-score of 89.82, close to the best value found, 91.66, for the LSTM with a window size of 70. The methods performance over the increase of window size is represented in Figure 6.

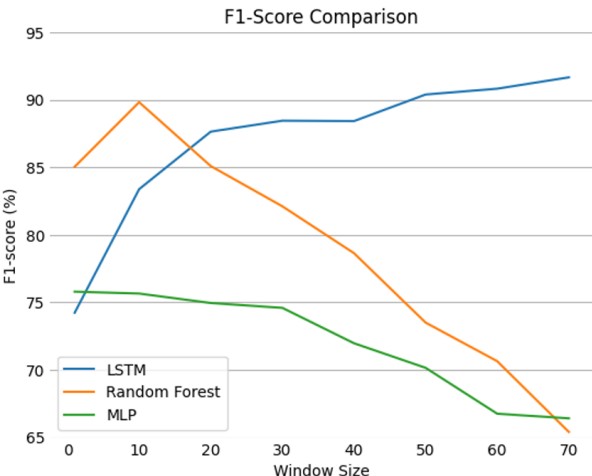

**Figure 6.** Performance over window size.

The best performing models are LSTM-70 and RF-10, and they share the same value of FPR (Table 3). However, the LSTM presents a higher precision and lower recall in comparison with the RF. Since the values of these metrics are more balanced for the LSTM 94.03 precision and 89.71 recall, it results in the highest F1-score, 91.66. The complete results are presented in Table 3.

**Table 3.** Results for the multi-flow approach.

| Model | Accuracy | Precision | Recall | F1-Score | FPR |
| --- | --- | --- | --- | --- | --- |
| LSTM-70 | 99.94 | 94.03 | 89.71 | 91.66 | 00.04 |
| RF-10 | 99.95 | 96.83 | 85.65 | 89.82 | 00.04 |

These results lead to believe that the multi-flow approach outperforms the single-flow-based one and that the LSTM is a robust algorithm for understanding complex patterns in sequential data, in particularly, network traffic data. Furthermore, the algorithms performance seems to keep improving as the window size grows larger. Optimizing the value of the window can be a crucial point for obtaining the best possible intrusion detection classifier for the CIDDS-001 context.

### 3.1.3. SATIE Toolkit Preliminary Results

The normal usage of the SATIE Toolkit and the scenario simulation runs produced, on a regular basis, several Alerts and Incidents. These data, although not being the best to serve as testbed for ML models, were used to obtain some preliminary results for the Incident probability algorithm. These experiments were essential to understand which approaches are better for the SATIE data and how well can the algorithms distinguish between malicious Alerts, which were tagged as Incidents, and false positive Alerts. The considered dataset was built with data extracted from the Investigation Database, which was in turn obtained by the Synchronization Mechanism continuous execution. All the Alerts related to Incidents, 368, were labelled as malicious while the remaining ones, 9215, were marked as normal. The dataset is not large in terms of data volume and has a high-class imbalance since more than 96% of records are benign. These characteristics made the

application of deep learning approaches such as MLP and LSTM unviable. Additionally, there were multiple challenges regarding data quality such as Alerts related to Incidents that were not manually labelled in the IMP, Alerts with a lot of empty fields that were only generated to test SATIE Tools and many repeated entries due to simulations that are executed daily. To mitigate these problems, every feature with over 60% missing values were discarded as well as all the Alerts related to the repeated daily executions. Furthermore, an oversampling method, Synthetic Minority Oversampling Technique (SMOTE) was used to produce synthetic examples of Incidents to minimize the class imbalance.

The data, after being pre-processed, was split into two sets: 70% for training and 30% for the test. Then, a RF model was used as a classifier (RF-1), obtaining an accuracy of 98.08%. However, the value of F1-score, 60.94%, indicated that the model was performing poorly on the minority class, failing to classify most of the Incidents. In an attempt to improve the obtained results, three time-based features were engineered for a given window of time (30 min): the number of Alerts, the number of distinct sensors and the most common sensor. With these new features, the accuracy and F1-score of this new classifier (RF-2) improved significantly, 98.54% and 76.60% respectively. The preliminary results lead us to believe that an approach which combines both individual Alert features and time-based engineered features can work quite well on the SATIE data. On the other hand, the dataset extracted from the Correlation Engine, despite its limitations, was a good starting point to fine tune the SMS-I ML algorithms. This was improved using the different scenario simulation executions that were executed on the platform, learning new patterns that was used to identify Incidents more accurately in the demonstration phase.

### 3.2. Association Rule Mining

Apriori is a very popular algorithm for data mining focusing on association rules, developed by Agrawal and Srikan in 1994 [35]. It identifies the items or patterns in a transactional dataset and then relates frequent occurrences to those patterns, generating association rules to describe them [36]. These rules are comprised of statements that describe the relationships between seemingly unrelated items inside a transaction.

Let $X = \{i_1, 1_2, \ldots, i_m\}$ be the set of all items concerned in a dataset, and $T = \{t_1, t_2, \ldots, t_m\}$ be a set of transactions, where each transaction is a set of items. The association rule, noted as $X \Rightarrow Y$ indicates a certain relation between two itemsets $X$ and $Y$. An association rule $X \Rightarrow Y$ is supported if the percentage of transactions that contain both itemset $X$ and $Y$ in $T$ exceeds a certain threshold, called support threshold, i.e., $\texttt{Support}(X \Rightarrow Y) = \frac{\text{Number of transactions containing } X \text{ and } Y}{\text{Total number of transactions}}$. Furthermore, the confidence for the association rule $X \Rightarrow Y$ is defined by the percentage of transactions that contain itemset $Y$ among transactions containing itemset $X$, i.e., $\texttt{Confidence}(X \Rightarrow Y) = \frac{\text{Number of transactions containing } X \text{ and } Y}{\text{Number of transactions containing } X}$. The support represents the usefulness of the discovered rule and the confidence represents certainty of the rule. Lift is a simple correlation measuring whether $X$ and $Y$ are independent or dependent and correlated Events. It is calculated by $\texttt{Lift}(X \Rightarrow Y) = \frac{\text{Number of transactions containing } X \text{ and } Y / \text{Number of transactions containing } X}{\text{Percentage of transactions containing } Y}$. If a rule has a lift of one, $X$ and $Y$ are independent and no rule will be generated containing either event. If a rule has a lift greater than one, $X$ and $Y$ are dependent and correlated positively.

To build the association rule mining for the SMS-I tool, using the apriori algorithm, the sequences of Alerts in a mineable database were grouped by using a certain criterion to form transactions. That criterion is a time window, and the focus will be the name of the sensor that originated the Alert. In order to compile the transactional dataset, for each Alert, the selected window was subtracted to its "detect_date" field. From the obtained time range, all Alerts that fell inside that interval were joined and a list with their sensor's name was created, performing this operation on all entries, and obtaining the set of transactions. Using this set of transactions several rules are generated to allow the user to understand the correlation of the different sensor Alerts in an attack.

### 3.3. Incident Response

After positive identification of an Alert as a security threat, measures need to be taken to limit the impact of the attack. These mitigation measures are usually described in procedures that detail, step-by-step, how to proceed when dealing with a given type of attack. These procedures are then compiled into playbooks [37] that can be extended to perform other important tasks in the mitigation or remediation process, tailored to the organization that is using them. A "phishing email" playbook, for example, might not only include the normal steps of deleting emails from affected inboxes and running scans on the machines of victims to make sure nothing was compromised, but also send out personalized memos raising awareness about this type of attack.

Compiling a list of playbooks for different types of attacks allows automation of much of their steps, considerably streamlining a SOCs workflow [37]. Additionally, multiple Alerts originating from the same type of attack, or even the same attack, can be aggregated in cases where playbooks can be applied to all the Alerts in a case at the same time.

In order to further automate cybersecurity, and focusing on Incident Response in a SOC workflow, a new module of SMS-I tool (Incident Response) was designed to be capable of slotting into current SOC tools, as a way of enriching incoming Alerts. This module is leveraged as a decision support system, employing multiple models to perform identification and classification of Alerts, adding their results as another point of consideration for security expert analysis. The additional information helps analysts not only decide if a given Alert is in fact an attack, but also by identifying which case contains playbooks to treat similar Alerts.

SMS-I Incident Response module aims to tackle two problems of the SOC pipeline: classification of incoming Alerts for security threats (Alert Classification); and grouping of similar Alerts in cases for bulk processing (Alert Aggregation). Although different in their nature, both of these are classification problems where a set of data points are categorized into classes. In this context, the data points will be Alerts and the classes their possible label.

In the Alert Classification problem, only two possible classes exist for an Alert, either attack or normal. In contrast, in the Attack Aggregation problem, the possible classes are the existing cases in the system. Furthermore, the nature of the data for Attack Aggregation binary classification problem guarantees that all the future incoming entries will only ever be of two possible types.On the other hand, classifying each Alert into groups will fail when a never before seen Alert, i.e., from a new type of attack, arrives in the queue. In this case, the multiclass classification model, trained with known classes will incorrectly identify the new Alert as one of the existing classes. For this reason, a middle step needs to exist between both classification problems—Attack Identification. After being classified as an attack by the first model, the system needs to decide if this Alert is similar to other Alerts already in the database or if it is a new one. As such, an anomaly detection model will be trained with Alerts already in the system to create a baseline of known Alerts, filtering any outliers and skipping the final step. The third model is trained on groups of Alerts that compose a case, selecting the relevant case for every incoming entry. The sequence of these three steps can be seen in Figure 7.

Each of the three different phases of the SMS-I Incident Response module, requires ML models tuned to the unique specifications of their given problem. These models will undergo a selection stage where data originating from the final system is used to train and compare the results among them.

#### 3.3.1. Alert Classification

The first step in this Incident Response pipeline will analyse an Alert in order to classify it as an attack, or not attack. If the Alert receives the "not attack" classification, then the Alert is the result of a false positive and can be safely disregarded. On the other hand, if the Alert is considered an attack, it will continue to the next step of this pipeline. This binary classification problem is extensively studied in this domain, with multiple models

continually being researched in the literature [38–40]. Three models were selected for this first step:

- **Random Forest**, as already mentioned, is a tree based model, employing a set of decision trees and taking in account the output of each one. A decision tree aggregates datapoints by iteratively splitting the features of a given dataset into consecutive binary nodes, ending each branch on its outcome, or label. Although very good with low complexity data, higher sized trees can lead to overfitting. Random Forest models mitigate this issue by using an ensemble of unrelated decision trees and consolidating their results, achieving significant results in the literature for both classification and regression problems.
- **Support-Vector Machines (SVM)** [41] is a probabilistic model that maps training data to points in space, and finds the hyperplane with the maximum margin that separates the two classes. Newer data points are mapped in space in the same way and classified according to which side of the hyperplane they have landed. This model is a very robust classifier with the caveat that it is limited to binary-class classification.
- Similarly to Random Forest, **XgBoost** [42] is an ensemble of decision trees, but using a gradient boosting algorithm. Instead of concurrently training a group of decision tree models and averaging their output, models are trained consecutively using the residuals from each iteration to train the next one.

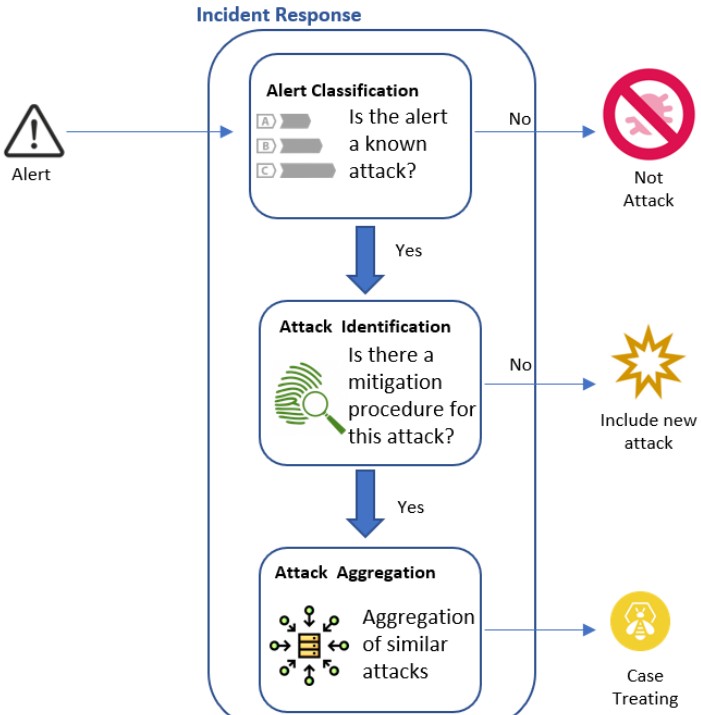

**Figure 7.** SMS-I Incident Response module Architecture.

### 3.3.2. Attack Identification

In order to classify an incoming Alert as "unknown", an anomaly detection based approach was selected. Although this approach is not uncommon for the cybersecurity domain, it is normally applied to the detection of attacks, whereas here, it is used to identify Alerts different from everything in the system.

After classification in the first step, an Alert classified as attack is analysed for known information. The objective is to aggregate this new Alert with other Alerts in the system. If the incoming Alert is known, it will be assigned to a case containing playbooks on how to deal with this type of attack. If it is unknown, the Alert is marked as such, to be analysed

and procedures prepared on how to deal with this type of attack. For this novel use case, two models were picked from the literature, as the most suitable:

- **Isolation Forest** [43] is a tree based model that uses distance between data points to detect outliers, hinging on the principle that outliers are distinct from normal data. During the construction of the binary tree, data are grouped into branches according to their similarity, with more similar entries needing longer branches to differentiate them. As such, data closer to the root of the tree can be considered an anomaly since it was easily distinguishable from the rest.
- **One-Class Support-Vector Machines** [44] is a similar implementation to SVM but instead of using an hyperplane to separate two classes, it uses an hypersphere around normal data and classifies new data based on its distance to the sphere.

### 3.3.3. Attack Aggregation

Finally, in the third step, Alerts previously marked as both "attack" and "known" in previous steps are matched to the Alerts in the system, searching for a suitable case to be assigned to, allowing automatic application of remediation or mitigation techniques contained in the related playbook.

The multiple possible results for this step, cases, makes this a multiclass classification problem, a subset of normal classification. As such, some models from the first step were also selected:

- **Random Forest** due to its robust results and straightforward implementation, behaving no differently in binary and multiclass classification problems.
- Although models such as **Support-Vector Machines** in its most simple type only supports binary classification, implementations exist where the problem is compartmentalized into multiple binary classification problems followed by the same principle: discovering the hyperplane that linearly separates classes [45,46].
- **K-Nearest Neighbors (KNN)** [47] uses distance between datapoints to identify clusters of similar data. Despite its good results it is not very scalable due to being computationally demanding.

### 3.4. Preliminary Results

Despite our first evaluation of which models should be used for each phase of the SMS-I Incident Response module, we need to test them in a dataset to select the one that should be deployed. For that, we used the testbed dataset already described in Section 3.1.1. In Table 4, we present the results for each phase. Note that we only consider the models described in the previous section, because they have already been chosen as the best approaches to be tested.

**Table 4.** Incident Response Experiment Results.

| Steps | Models | Accuracy | F1-Score | Macro F1-Score |
|---|---|---|---|---|
| Alert Classification | RF | 97.1 | 69.2 | 96.8 |
| | SVM | 97.3 | 63.1 | 96.5 |
| | XgBoost | 97.3 | 70.4 | 96.9 |
| Attack Identification | IF | 80.9 | 82.8 | 80.8 |
| | One Class SVM | 67.6 | 73.7 | 65.6 |
| Attack Aggregation | RF | 80.2 | 58.5 | 77.8 |
| | SVM | 80.2 | 59.2 | 78.3 |
| | KNN | 88.3 | 54.9 | 85.4 |

F1-score was selected as the metric of choice given its good balance between Precision and Recall while paying attention to class imbalance existent in the data. This imbalance can also be observed in the difference between macro and weighted metrics, since macro metrics take into account the number of each class' members during result calculations. As such, for the first and third steps, the macro F1-score was used to evaluate the impact differently sized classes have in the final results. For the second phase F1-score was also used, only this time focusing on the score for the outlier class, i.e., the Alerts considered unknown to the system. For the first and second step of the SMS-I Incident Management module, tree based models achieved the best performance in the experiments, with XgBoost and Isolation Forest respectively selected for the mentioned steps. For the third step's experiments, although a mostly inconclusive affair due to the closeness of results, SVM did manage to edge out ahead.

## 4. SMS-I Intelligent Dashboard

SMS-I allows the analysis of data from heterogeneous systems over different time frames. To provide this information regarding the system's Events, Alerts, and Incidents in a useful way, it implements a visualization tool—the SMS-I Intelligent Dashboard. Furthermore, it assists and facilitates the security analyst's work using graphical dashboards and Alert classification suggestions, which derive from the SMS-I ML Engine previously presented. Consequently, users can more easily identify anomalous situations that can be related to possible Incident occurrences. They can also explore information, with varying levels of detail, including logical information and technical specifications. An overview of the different information provided can be seen in Figure 8.

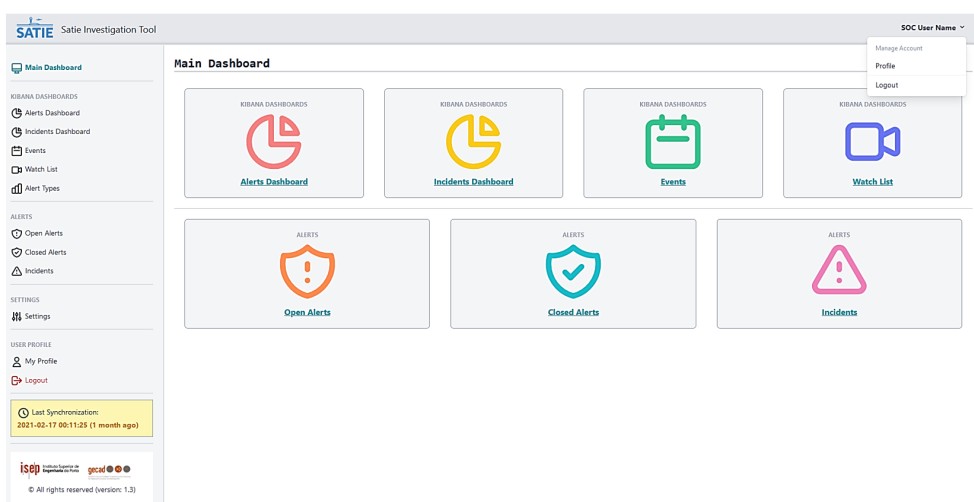

**Figure 8.** SMS-I Intelligent Dashboard overview.

Two different detailed dashboards were accessible: Alerts and Incidents Dashboards. Both were developed using Elasticsearch and Kibana technologies. Elasticsearch is responsible for the analysis, normalization, enrichment and storage of Alert and Incident data, as well as data provided by ML algorithms. Then, these data are accessed by Kibana to create these two dashboards, which allow the user to search and visualize airport security related data.

The Alerts Dashboard includes all data related to airport security Alerts generated by the different cyber and physical Threat Detection Systems available in the SATIE Toolkit. One of the main goals of this dashboard is to monitor the quantity, nature, and severity of Alerts, considering their Incident prediction probability, which is calculated by the SMS-I ML Engine. More than 70% of security analysts feel overwhelmed with the number of Alerts and Incidents they need to investigate for a day [48]. In addition, more than 50% of organizations receive over 10,000 Alerts daily, which can lead to Alert fatigue and

neglect. Therefore, to maintain SOC efficiency and reduce the impact of the investigation on the responsible personnel, it is essential to control the quantity of received Alerts and Incidents. Therefore, a set of graphics and metrics were added to this dashboard (see Figure 9) to monitor the number of Alerts received to help avoid a sudden overload of Alerts by monitoring the total number of cyber and physical Alerts. In addition, an Alert gauge was added to ensure that an overwhelming quantity of Alerts is not reached.

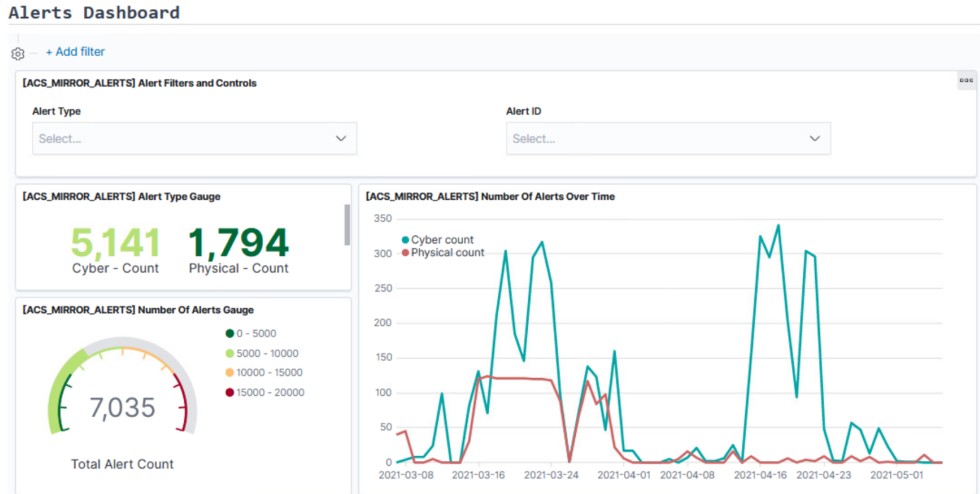

**Figure 9.** SMS-I Intelligent Dashboard: Alert quantity monitoring visualizations.

The severity of Alerts is another important parameter that needs to be monitored by security analysts, since Alert's severity defines if the Alert should be ignored or if there is a need to conduct a more thorough investigation. For the SATIE project, four severity levels were defined: high, medium, low, and info. Besides controlling the number of Alerts for each severity level, to avoid the overburdening of security analysts, using the Alerts dashboard is also possible to monitor the date of occurrence of Alerts (see Figure 10). This is useful to perform pattern and trend identification and to study previous Incidents and preceding Alerts.

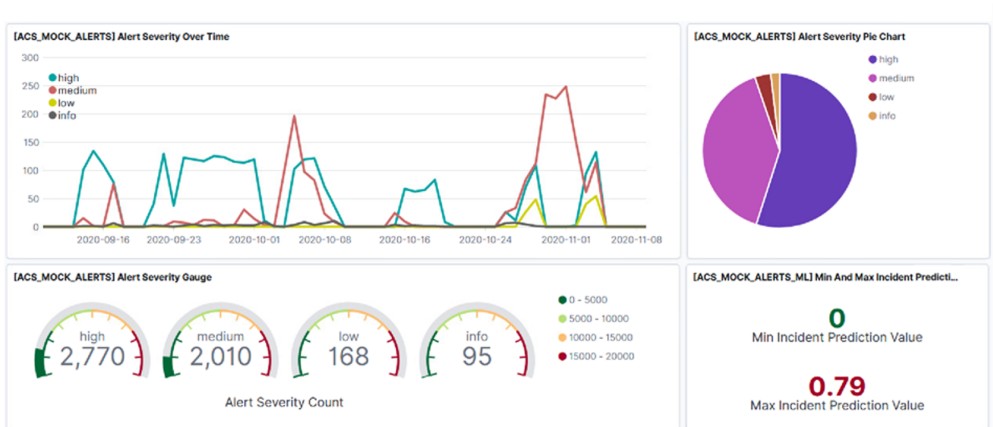

**Figure 10.** Alerts Dashboard—Alert severity monitoring visualizations.

The results provided by the ML engine regarding the Incident prediction probability, in other words, the probability of an Alert representing an Incident, can also be visualized in the Alerts dashboard (Figure 11). A set of graphics and metrics display, from 0% to 100%, the number of Alerts that possess a certain probability of being an Incident, as well as the average Incident prediction probability. In the example shown, most Alerts have an

Incident prediction probability lower than 35%, which leads to a low average probability value. This means that overall, there probably is not an occurrence of an Incident.

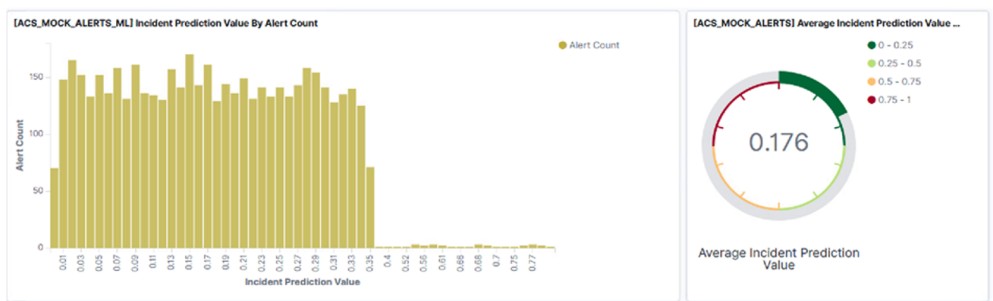

**Figure 11.** Alerts Dashboard—Incident prediction probability visualizations.

The most common source and target IPs and ports are also displayed to the user in the Alerts Dashboard (Figure 12). This information can be very valuable for the security analyst, as it helps to discover information about the attacks, namely where they come from and what the targets are.

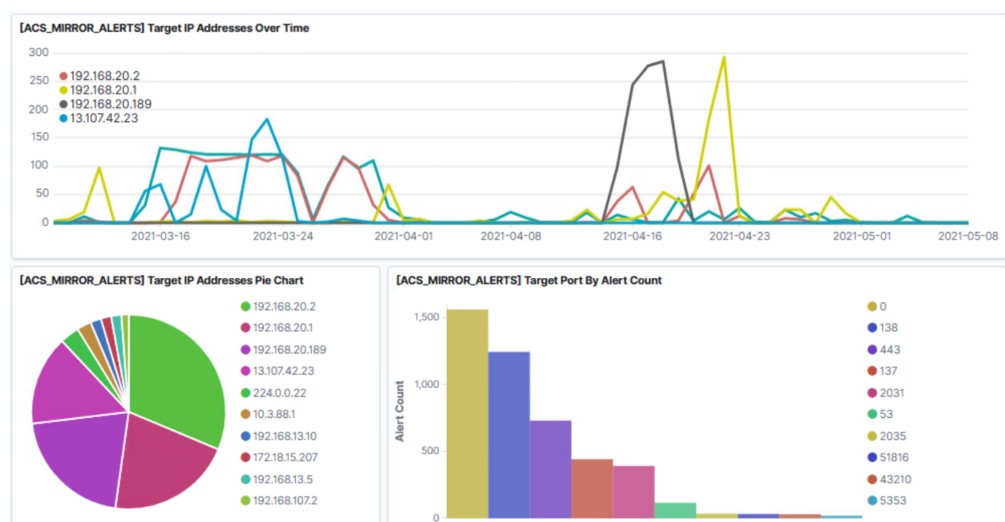

**Figure 12.** Alerts Dashboard—Target IP and Ports visualizations. Note that IPs have been obfuscated for security reasons.

The Incidents Dashboard aggregates all detected Incidents related to airport security. This dashboard follows the structure of the Alerts Dashboard by monitoring the quantity, nature, and severity of Incidents (Figure 13). Thus, similar to what happens with the Alerts Dashboard, it has similar visualizations available to the user, displaying information regarding Incident quantity monitoring and Incident severity monitoring.

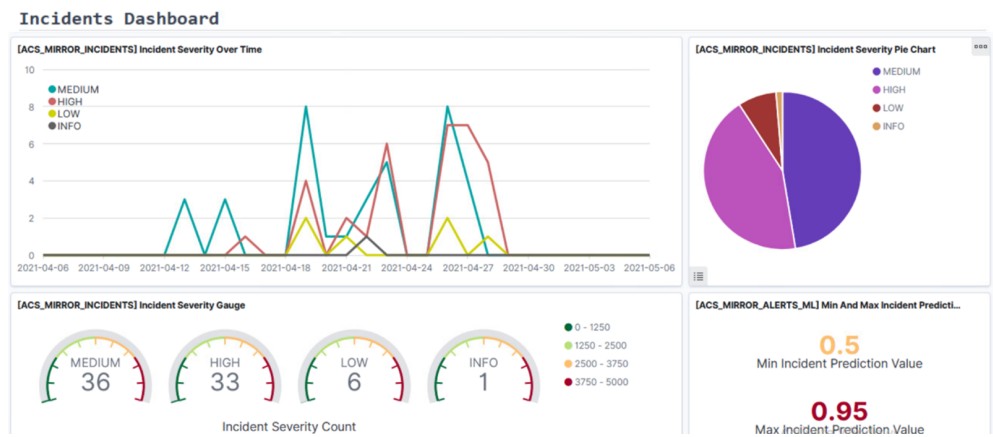

**Figure 13.** Incidents Dashboard—Incidents severity monitoring visualizations.

SMS-I Intelligent Dashboard also makes available a set of different visualizations. Events timeline is one of them. It provides ability to security analysts to preview a timeline of Events within the system. Events are displayed in the form of an ordered timeline, with summarized info of each event (Figure 14). Filters can be applied to customize the timeline, such as: maximum Alerts number, minimum Incident probability, and time range.

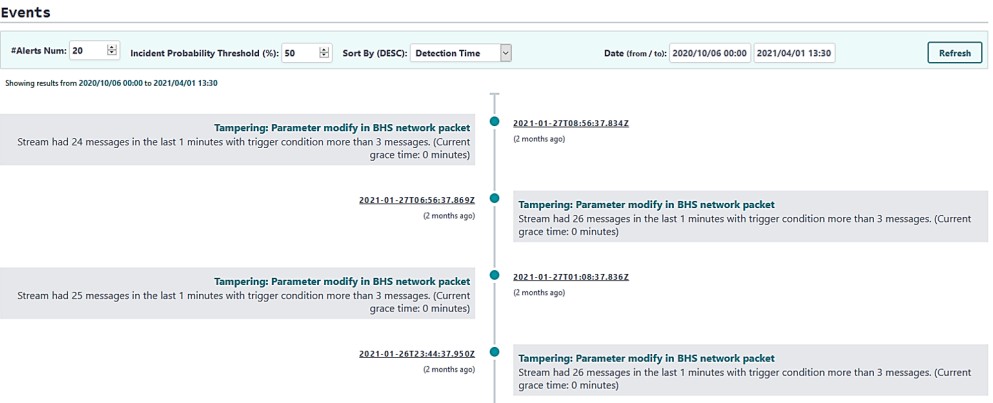

**Figure 14.** SMS-I Intelligent Dashboard: Events timeline.

A Watch List section is also available and allows users to preview a list of the latest Alerts within the system (Figure 15). Alerts in this list are being displayed in the form of aligned cards, with summarized info of each Alert within the corresponding card. The list can be sorted by detection time or Incident probability, and filtered by maximum Alerts number, minimum Incident probability, and time range.

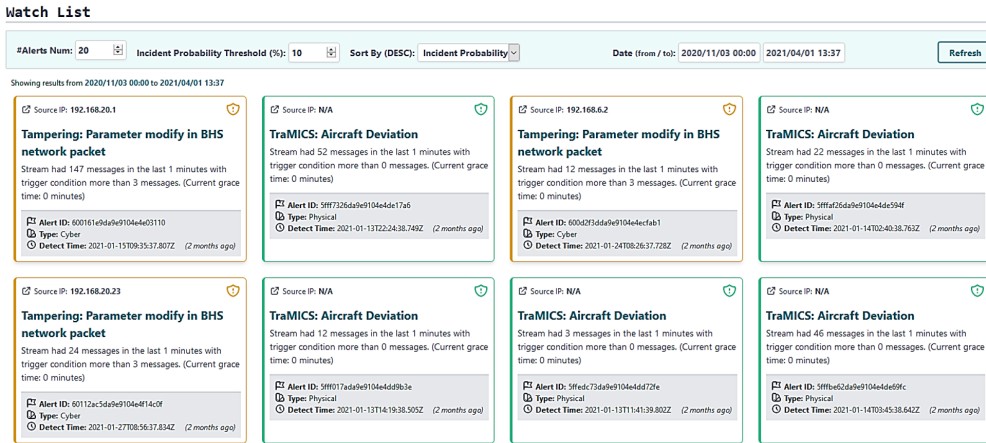

**Figure 15.** Watch List with example Alerts.

Each card within the list has highlights of the Alert details (Figure 16). Users can click on any card to display the full details of the corresponding Alert (Figure 17). Furthermore, cards are displayed using indexed colours that reflect the severity level of each Alert (red for High, orange for Medium, and Green for low).

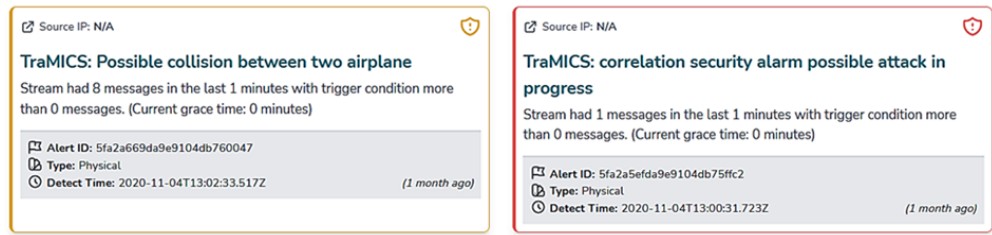

**Figure 16.** Watch List Alert Cards example (see [49] for more information on TraMICS).

When the user clicks on a specific Alert Card, the corresponding Alert details will be displayed. Details include the Alert title and description, information identifying the Alert, the source and target details, and the probability of this Alert being an Incident.

If the card is a specific Incident Card, the corresponding Incident details as well as the related Alerts will be displayed (Figure 18).

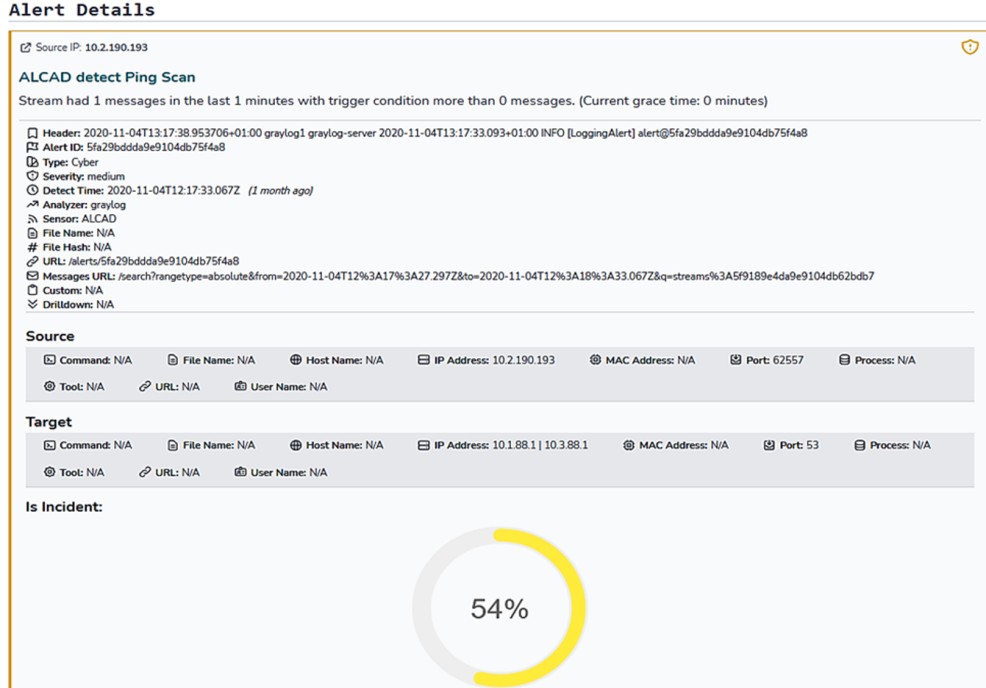

**Figure 17.** Alert Details example.

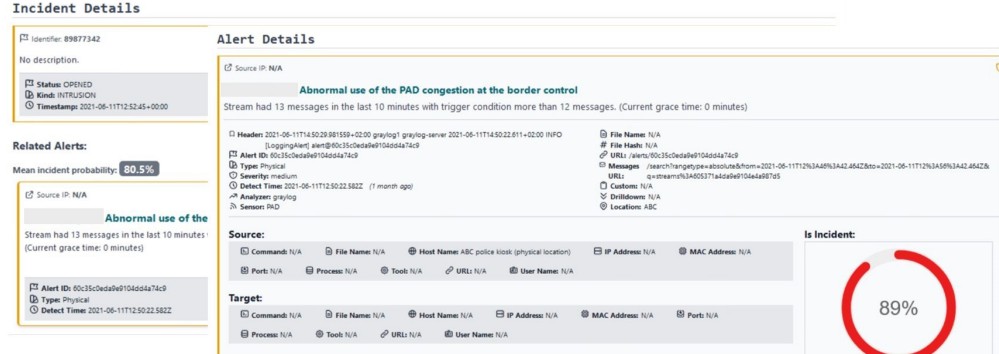

**Figure 18.** SMS-I Intelligent Dashboard: Incident and Alert details example.

It is also possible to display the distribution of Alerts as per their types (physical/cyber), and due to multiple levels of aggregation (no aggregation, by minutes, by hours, by days, . . . ), using the Alert Types section of SMS-I Dashboard (Figure 19). Alerts can be also filtered by their type, Incident probability, and detection time.

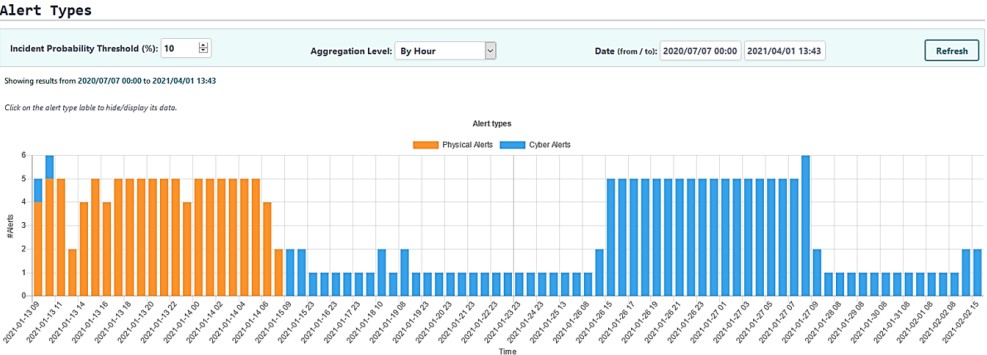

**Figure 19.** Alert Types visualization.

Another important part of SMS-I Intelligent Dashboard is the Association Rules functionality (Figure 20) which allows security analysts to automatically generate rules that can help them understand, using historical data, the correlation of the different sensor Alerts in an attack. The security analyst can customize the parameters, namely the time window, the support and confidence, to generate different rules.

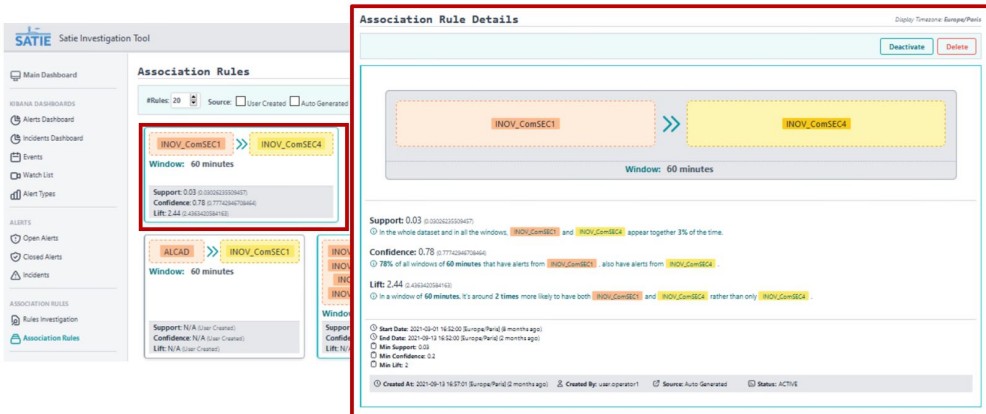

**Figure 20.** SMS-I Intelligent Dashboard: Association rules visualization.

## 5. SMS-I Incident Response Integration

TheHive is an Incident management tool focused on Incident analysis used by security analysts to manage Incidents and give them an adequate treatment. This tool is often used by the organizations due to its open-source implementation, and collaboration focused functionalities. TheHive is designed to support multi-enterprise SOCs in a collaborative Incident management and orchestration environment. This allows security analysts and experts to share information between partners and work on cases collaboratively. Furthermore, TheHive contains connections to security threat databases, namely MISP, receiving up-to-date intelligence on any new security threats.

SMS-I allows a direct integration with the TheHive tool. Due to TheHive's highly collaboration focused functionalities, this integration can be described as in Figure 21, with the novel SMS-I Incident Response module capturing incoming Alerts from multiple sources and, after ML analysis, augmenting their information with intelligent classification.

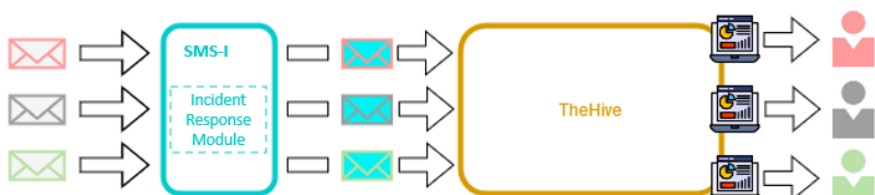

**Figure 21.** SMS-I Incident Response module information flow.

The improved Alerts are then submitted to TheHive's new Alert queue, waiting for manual verification (Figure 22). When security analysts log in to TheHive to perform this verification, they can use the ML analysis contained in each Alert to help decide on how to proceed with each one. This information is very useful, since the security analyst does not need to try to understand if there already exist similar attacks in the database, for example. This information is already provided by SMS-I in the additional fields of the Alert (Figure 23).

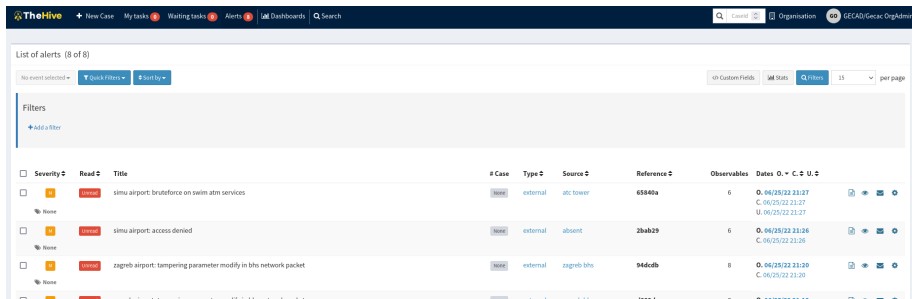

**Figure 22.** TheHive Alert Queue.

TheHive utilizes its own concept of observables [50], stateful properties of an Alert that are likely to indicate an intrusion, allowing investigations to be run on individual or groups of observables to verify their compromise level (Figure 23). Therefore, source IP, file hash, or sender email domain are fields contained in an Alert received by SMS-I that can be considered observables. In the scope of TheHive, this is information that may indicate an attack. Since SMS-I already provides this information, the security analyst does not need to manually add it.

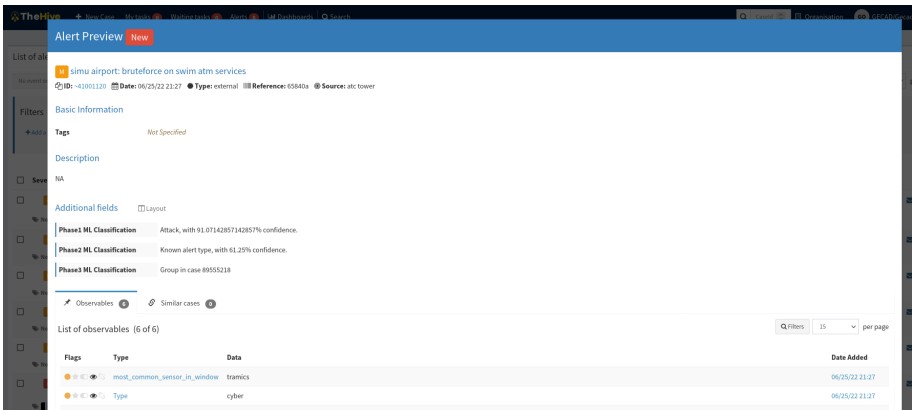

**Figure 23.** TheHive Alert Example.

## 6. SMS-I Demonstration

SMS-I tool was validated and demonstrated in the scope of the SATIE project, using a simulation platform and in the pilot sites [51–53]. The different security analysts were first introduced to the platform. First, we explained the purpose of the SMS-I tool as a whole, and then we showed how they can get useful insights from the information in the SMS-I Intelligent Dashboard. Then, the security analysts used the SMS-I tool through the SMS-I Intelligent Dashboard. During the simulations and then in the demonstrations several data and opinions were gathered and used to fine tune the tool and refine the SMS-I ML engine. All the experiments also highlighted the need to have tools such as SMS-I, that intelligently correlate the different cyber and physical security Alerts and assist the security analysts to detect highly sophisticated attacks of this time and the future. IBM stated that it took an average of 287 days to identify and contain a data breach in 2020 [54]. This detection time demonstrates how difficult is for companies to detect and mitigate cyber attacks [55]. This is even more difficult in CPS, where attacks usually involve multistage and multiple components. Moreover, the analytic tasks conducted by security analysts rely heavily on a cognitive decision-making process that can differ between analysts, depending on their technical knowledge or level of experience [56]. This is why it is so important to have intelligent tools, as SMS-I, to support security analyst decisions.

To demonstrate the efficiency of the different tools in the SATIE toolkit several realistic scenarios incorporating a considerable number of potential cyber and physical attacks were defined. In one of these threat scenarios, an attacker seeks to perform cyber attacks

on the Airport Operation Control Center (AOCC) system to manipulate the information displayed in the Flight Information Display System (FIDS), thus giving origin to passenger movements which result in an irregular and disorderly movement of people in the terminal, and odd plane movements on the platform to create confusion on the apron. The attacker's first actions can be used to demonstrate the effectiveness of the SMS-I tool and the help it can give to security analysts in their decision-making process. The scenario starts with an attacker who sends a spear-phishing email to a computer with administrator privileges. An employee opens the email on that computer and clicks on the link which allows the malware to be downloaded and executed. This malware allows the attacker to take remote control of the computer. Then, the attacker performs a network scan to determine the network address and port of the Airport Operation Database server—his main target. From a security analyst's perspective, it is important to correlate both Events and understand that they are steps of the same multi-step attack. However, due to the difficulty of analyzing these different Events, which can be, for example, observed and classified by different analysts, they are sometimes classified as isolated Events instead of being correlated and aggregated. This was what happened in the demonstration of this scenario. The security operator reported the corresponding Alerts as two different Incidents, as can be seen in Figure 24.

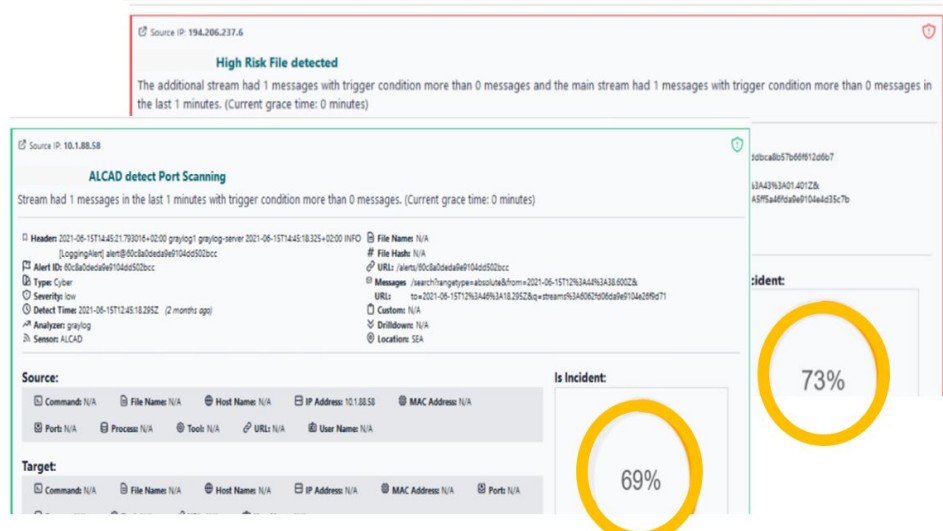

**Figure 24.** SMS-I Intelligent Dashboard: Malware Detection by Malware Analyser and Network Scan detection by ALCAD system (part of SATIE Toolkit) [57].

Moreover, the port scanning Alert was classified as a low severity Incident, which should not be the case since it is already the second step of the multi-step attack.

Using the SMS-I Intelligent Dashboard, after the reporting of the Incident by the security operator, the security expert can observe that. Despite this being an Incident that was reported as a low severity Incident, it is related to an Alert that has a 69% probability of being an Incident (Figure 24), thus it should be reported with higher severity. Similarly, the SMS-I Incident Response module classifies the port scan Alert as an attack with 66% confidence (Figure 25), while not discovering similar Alerts in the system. This means a playbook should be created with steps mitigating this type of attack so that future attacks of the same type can more easily be treated.

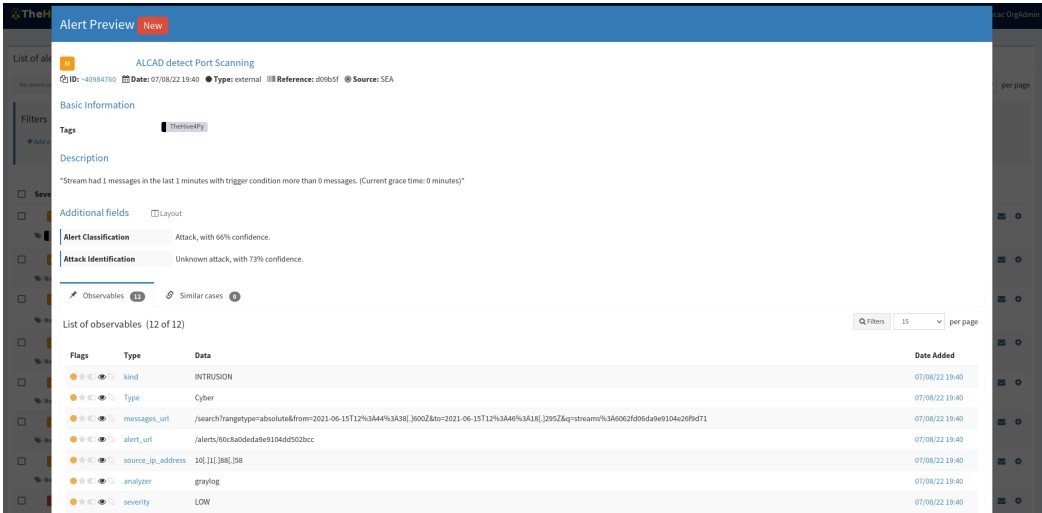

**Figure 25.** SMS-I Incident Response results for Network Scan example (TheHive view).

Furthermore, using association rules, the security analyst can understand that the malware and the network scan Alerts are correlated and should be reported as being part of the same Incident (Figure 26). This information can also be added to the playbook to have more information about this type of attacks.

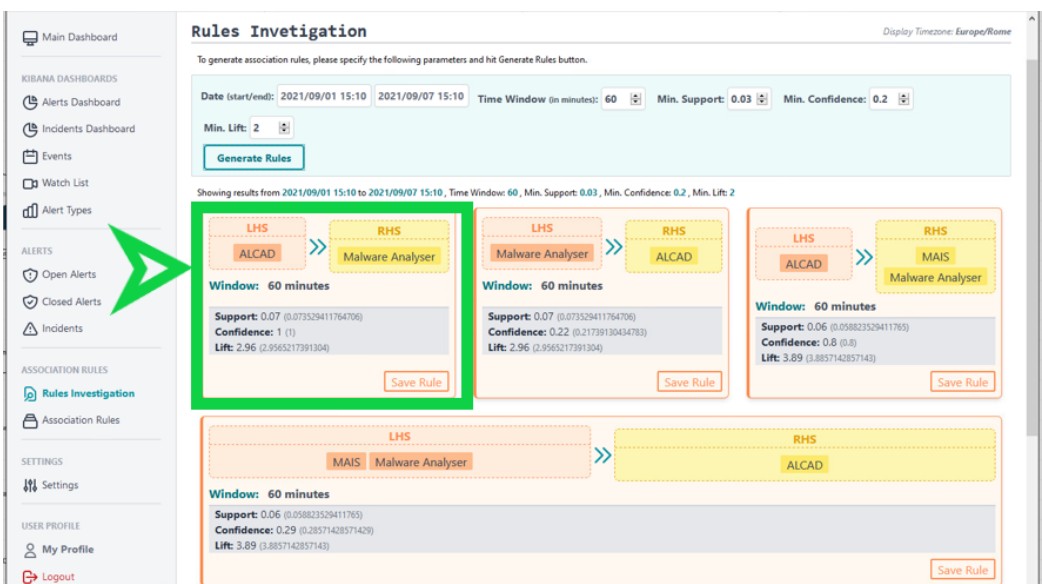

**Figure 26.** Association Rules generated by SMS-I tool. The rule marked is the one generated by the example described.

Therefore, with this demonstration, we showed not only the ability of the SMS-I to support the security experts work, but it also allowed us, using a very simple "real" example, to illustrate the need for intelligent tools that can assist security analysts in their decision-making process. Using the SMS-I tool, the security analyst can understand the weaknesses of the first security analysis and have intelligent suggestions on how to combat and even resolve them. Different suggestions are provided to the analyst to define mitigation measures to avoid future attacks. Furthermore, SMS-I also simplifies the sharing of information, through TheHive platform, to support the security awareness of other partners.

## 7. Conclusions

This work describes the SMS-I tool that allows the improvement of the forensics investigation in cyber–physical systems. It is a complex system composed by multiple components with specific functions, namely periodic data synchronization, Incident prediction and response, association rule mining, dashboard visualization, and a several other functionalities involving different lists and filters.

Several AI approaches were used to process and analyse the multi-dimensional data exploring the temporal correlation between cyber and physical Alerts. Supervised algorithms were trained on the sequential data of cyber and physical Alerts to predict the probability of a given Alert to be an Incident based on previous occurrences. The results obtained suggest that the multi-flow approach outperforms the single-flow-based one and that the LSTM is a robust algorithm to understand complex patterns in sequential data, in particularly, network traffic data. Forest-based models achieved the best performance in all tasks considering Incident response analysis. In addition, several association rules can be created by applying different ML techniques that allows the user to understand the correlation of the different data in an attack.

All the information can be visualized in the SMS-I Intelligent Dashboard. Several graphical dashboards, with different levels of detail can be used to easily identify anomalous situations that can be related to possible Incident occurrences. Furthermore, the information provided by the ML algorithms, namely the Incident probability can be analysed on SMS-I intelligent dashboard. Moreover, for an additional insight about the association rules, a management of the association rules by the security analysts can also be done.

The integration between SMS-I tool and TheHive, an Incident management tool, was presented. This integration supports the collaboration among the security professionals, not only inside the same institution but also between companies. Furthermore, SMS-I provides an extra intelligent layer that adds useful information to the security occurrences, which is automatically displayed in the Incident management tool facilitating information sharing and improving the quality of the investigation.

SMS-I tool was tested in different European airports in the scope of SATIE project. A very simple and authentic example, presented in this work, demonstrated the convenience and usefulness of the SMS-I tool in the decision-making process of security analysts. As future work, we plan to test SMS-I in other cyber–physical systems to improve the results across the board. On the system's side, a greatest improvement could be an automatic retraining of the models, using labeled data from the SOC.

**Author Contributions:** Conceptualization, E.M., N.S., N.O., S.W., O.S. and I.P.; methodology, E.M., N.S., N.O., S.W., O.S. and I.P.; software, N.S., N.O. and S.W.; validation, E.M., N.S., N.O., S.W., O.S. and I.P.; formal analysis, E.M., N.S., N.O., S.W., O.S. and I.P.; investigation, E.M., N.S., N.O., S.W., O.S. and I.P.; writing—original draft preparation, E.M., N.S., N.O. and S.W.; writing—review and editing, E.M., O.S. and I.P.; visualization, N.S., N.O. and S.W.; supervision, E.M., O.S. and I.P.; project administration, E.M. and I.P.; funding acquisition, I.P. All authors have read and agreed to the published version of the manuscript.

**Funding:** This research was funded by the Horizon 2020 Framework Programme under grant agreement No 832969. This output reflects the views only of the author(s), and the European Union cannot be held responsible for any use which may be made of the information contained therein. For more information on the project see: http://satie-h2020.eu/.

**Institutional Review Board Statement:** Not applicable.

**Informed Consent Statement:** Not applicable.

**Conflicts of Interest:** The authors declare no conflict of interest.

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
