# Peer review of "SMS-I: Intelligent Security for Cyber–Physical Systems"

_information, doi:10.3390/info13090403_

Round 1
Reviewer 1 Report
I would suggest some grammar/word choice editing for the final version.
I also suggest expanding on the definitions of the terms, possible via footnotes/endnotes, to clarify what the authors mean.
I also suggest expanding the discussion to more fully relate the implications of the findings for this system.
I like your paper!
Author Response
Dear Reviewer,
We appreciate the time and effort that you have dedicated to providing your valuable feedback on our manuscript. We are grateful for your insightful comments on our paper, and we are glad to read that you enjoy our paper. We have been able to incorporate changes to reflect the suggestions that you provided. We have highlighted the changes within the manuscript.
We are able to respond to any further questions and comments you may have.
Sincerely,
Eva Maia.
Reviewer 2 Report
The paper is well structured and interesting to the readers. I suggest a few language corrections and typo errors. Include major contributions to the paper before organization of the paper. A literature review may be improved. The authors can also refer to following papers.
10.1016.j.ijepes.2021.107718, 10.1016.j.promfig.2020.04.020, 10.1155/2022/1742772, etc.
Author Response
Dear Reviewer,
We appreciate the time and effort that you have dedicated to providing your valuable feedback on our manuscript. We are grateful for your insightful comments on our paper. Thanks a lot for the suggestions of works to include in the literature review. We have been able to incorporate changes to reflect the suggestions that you provided. We have highlighted the changes within the manuscript.
We are able to respond to any further questions and comments you may have.
Sincerely,
Eva Maia.
Reviewer 3 Report
- The paper presents the detailed SMS-I tool overview. Main disadvantages of the article and questions are:
- low quality/resolution of most figures
- the software features and flows are described, even integration in real airports. But the result still is not clear. What is the difference before and after the integration in SOC work performance/quality, the overall impact on airports, etc?
- the paper looks like a bit (some UML diagrams, etc.) extended version of [15] Maia, E.; Sousa, N.; Oliveira, N.; Wannous, S.; Pra¸ca, I. SMS-I: an Intelligent Correlation tool for Cyber-physical Systems. 725 DPSC2022 Proceedings, 2022. What the new result is presented in the paper?
- Some parts of the paper sound as advertisement (sections 2, 3.3.1), as textbook (3.1.2, 3.3.1 – 3.3.3) and/or as user guide (sections 4, 6).
- - “Cyber-physical” is mentioned in Introduction, once at the beginning of 3.1 section, and in Conclusions only. Paper title is not appropriate to the content.
Author Response
Dear Reviewer,
We appreciate the time and effort that you have dedicated to providing your valuable feedback on our manuscript. We are grateful for your insightful comments on our paper. We have been able to incorporate changes to reflect the suggestions that you provided. We have highlighted the changes within the manuscript.
Here is a point-by-point response to your comments and concerns.
- low quality/resolution of most figures
Thank you for pointing this out. We agree with this comment. Therefore, we tried to improve the quality of the images.
- the software features and flows are described, even integration in real airports. But the result still is not clear. What is the difference before and after the integration in SOC work performance/quality, the overall impact on airports, etc?
You have raised an important point here. However, we believe we highlight this in Section 6. We have tried to clarify this further by adding an additional conclusion to this Section:
“Therefore, with this demonstration, we showed not only the ability of the SMS-I to support the security experts work, but it also allowed us, using a very simple “real” example, to illustrate the need for intelligent tools that can assist security analysts in their decision-making process. Using the SMS-I tool, the security analyst can understand the weaknesses of the first security analysis and have intelligent suggestions on how to combat and even resolve them. Different suggestions are provided to the analyst to define mitigation measures to avoid future attacks. Furthermore, SMS-I also simplifies the sharing of information, through TheHive platform, to support the security awareness of other partners.”
- the paper looks like a bit (some UML diagrams, etc.) extended version of [15] Maia, E.; Sousa, N.; Oliveira, N.; Wannous, S.; Pra¸ca, I. SMS-I: an Intelligent Correlation tool for Cyber-physical Systems. 725 DPSC2022 Proceedings, 2022. What the new result is presented in the paper?
You are right, and we explain this in the text: “A first draft was presented at [22], and a more complete version of this draft was presented at [23]. This work shows in more detail the capabilities presented in the previous works, but also introduces a new capability: the Incident response.”
- Some parts of the paper sound as advertisement (sections 2, 3.3.1), as textbook (3.1.2, 3.3.1 – 3.3.3) and/or as user guide (sections 4, 6).
Thanks for your comment. In Section 2, we introduce the SMS-I architecture and briefly describe each component, so maybe that's why it looks like an advertisement. Section 3 presents this SMS-I ML engine in detail which is why it might sound like a textbook to you. Section 4 describes the SMS-I Smart Dashboard which is another important element of the SMS-I tool. We've tried to explain all its components, which is why it might sound like a user guide, but we believe this is important to understand why it can be so useful for security analysts. Section 6 briefly describes an example that shows the ability of SMS-I to support the security specialist's work.
- - “Cyber-physical” is mentioned in Introduction, once at the beginning of 3.1 section, and in Conclusions only. Paper title is not appropriate to the content.
Thanks for your comment. We decided not to repeat the term “cyber-physical” throughout the work. At the beginning of section 3.1, we explained that events, alerts, and incidents are cyber-physical. So, as we are always mentioning and working with these terms, we decided not to repeat “cyber-physical” throughout the document so as not to disturb the reader.
We are able to respond to any further questions and comments you may have.
Sincerely,
Eva Maia.
Reviewer 4 Report
Application examples need to be supplemented.
Author Response
Dear Reviewer,
We appreciate the time and effort that you have dedicated to providing your valuable feedback on our manuscript. We are grateful for your insightful comments on our paper. Regarding your comment saying, “Application examples need to be supplemented”, we believe that we have provided several examples throughout the article and especially in section 6 we present a realistic application that shows the ability of SMS-I to support the work of the security specialist. We have tried to improve this section to make the application of the SMS-I tool clearer. We highlight the changes in the manuscript.
We are able to respond to any further questions and comments you may have.
Sincerely,
Eva Maia.